# A Metasurface-Based LTC Polarization Converter with S-Shaped Split Ring Resonator Structure for Flexible Applications

**DOI:** 10.3390/s23146268

**Published:** 2023-07-10

**Authors:** Erfeng Li, Xue Jun Li, Boon-Chong Seet, Adnan Ghaffar, Aayush Aneja

**Affiliations:** Department of Electrical and Electronic Engineering, Auckland University of Technology, Auckland 1010, New Zealand

**Keywords:** metasurface, L2C polarization converter, split ring resonator, flexible

## Abstract

This paper presents a metasurface-based linear-to-circular polarization converter with a flexible structure for conformal and wearable applications. The converter consists of nested S- and C-shaped split ring resonators in the unit cell and can convert linearly polarized incident waves into left-handed circularly polarized ones at 12.4 GHz. Simulation results show that the proposed design has a high polarization conversion rate and efficiency at the operating frequency. Conformal tests are also conducted to evaluate the performance under curvature circumstances. A minor shift in the operating frequency is observed when the converter is applied on a sinusoidal wavy surface.

## 1. Introduction

The polarization is a key characteristic of an antenna because the electromagnetic wave propagation can behave significantly in the inherent polarization-sensitive materials, which affects the antenna performance correspondingly. Therefore, the polarization conversion of electromagnetic waves becomes significant for certain practical applications, such as linear-to-circular (L2C) polarization conversion [1,2]. Conventional polarization manipulation methods normally employ complex structures, such as birefringent crystalline liquids or crystals. Consequently, these converters normally come in large sizes and thicknesses, which makes them impractical to embed into compact systems. Additionally, conventional converters normally have low conversion efficiency [1].

In recent years, metamaterial has become a new solution to electromagnetic wave polarization conversion. Metamaterials are materials that are artificially crafted and demonstrate specific properties by manipulating their internal microstructure [3]. These properties normally cannot be found in natural materials. For example, it typically requires multiple stacks of material layers to build up a unique 3D structure so that the fascinating functionalities of metamaterials can be realized and tuned for different applications. This not only brings a significant number of challenges in fabrication but also leads to extensive losses and cost. Many metamaterials consist of a complex combination of metallic wires and geometrical units. This normally requires sophisticated fabrication and assembling technology [4,5].

As a 2D metamaterial which features thin films composed of individual elements, metasurfaces can overcome the aforementioned obstacles. Metasurface generally consists of a periodic or quasi-periodic planar array of subwavelength unit cells [5]. By employing different types of unit cell structures and manipulating their parameters, metasurfaces can affect the frequency, amplitude, phase, and bandwidth of the electromagnetic waves in desired ways. Another advantage of metasurfaces is the extremely small thickness if a proper dielectric material is selected. This makes metasurfaces suitable for compact integration with other systems [1,2].

In [6], an L2C converter based on substrate integrated waveguide cavity unit cell for 15–15.7 GHz was proposed. However, the feasibility of fabrication is high due to the structure of metallic via holes, which can also increase the fragility of the entire surface, given the number and density of the via holes. In [7], a dual-band circular polarization converter based on anisotropic metamaterial was proposed for 4.5 GHz and 7.9 GHz applications; however, it employed a three-layer structure, which increased the difficulty for fabrication.

A three-band metasurface based polarization converter was presented in [8]. The design has a double layer structure with an “L”-shaped periodic element. Depending on the different incident angles, the converter can convert the transverse magnetic (TM) polarized incident wave into RHCP reflection wave at 12.5 GHz, transverse electric (TE) polarized reflection wave at 18 GHz, and left-handed circular polarization (LHCP) reflection wave at 22.5 GHz. The result of the proposed design was very sensitive to the incident angle. A significant deviation could be observed when the incident angle had a minor change. In addition, the double FR4 layers added complexity to the structure and overall thickness. Another L2C polarization converter for broadband applications was reported in [9], which employed a “%”-shaped unit cell on one side of the dielectric substrate and a metal patch on the other side. The converter had high efficiency and conversion rate at the desired frequency. However, the incident angle dependency was high. Similar to the design in [8], the double metallic layer was another drawback of the design. The proposed metasurface that achieves linear cross-polarization, linear-to-circular, or circular-to-linear polarization over the wide frequency range of 5–37 GHz is presented in [10]. It consists of a coupled rectangular split ring resonator with different arm width and covers the X, C, Ku, K, and Ka bands. An AMC (artificial magnetic conductor) based high-gain wideband circularly polarized printed monopole antenna for unidirectional radiation pattern is proposed [11]. The proposed antenna consists of an AMC reflector, CPW feed, ground plane with rectangular slot on the left side, and asymmetrical ground plane on its right side. The asymmetrical ground plane is used to produce the circular polarization. By using the AMC layer, the gain of the proposed antenna is increased from 3.3 to 8.8 dBic. The terahertz (0.49–1.88 THz) polarization conversion based on metasurface double split ring resonator is presented [12]. The efficient polarization conversion is achieved with the help of the resonator. The proposed design consists of three layers: the top one is the resonator ring array, the middle one is the dielectric layer, and the bottom one is the metal plate. The thermally tunable linear-to circular (L2C) converter is proposed with the help of dielectric metasurface [13]. The proposed design consists of zirconium oxide microsphere resonators, active strontium titanate cladding, and flexible polyimide substrate. The device can be used in imaging and 6G wireless communication. An ultrawideband reflective metasurface converter is designed for linear-to-circular polarization [14,15]. The proposed design is circular pie-shaped and is grounded with copper sheet. The proposed metasurface can convert a y-incident polarized wave into an RHCP wave. The L2C exceeds 98% in the frequency range of 20–34 GHz. A high gain directional CPW feed UWB planer antenna with a new FSS unit cell is presented. It is Mercedes artistic-shaped and uses a circular ring with three straight legs for bandwidth enhancement. The FSS is used for the stopband filter to cover frequencies from 2.2 to 12.7 GHz. The bandwidth, gain, and efficiency of the proposed antenna are 136%, 11.5 dB, and 89%, respectively.

Through a careful literature review, a low-profile single-layered metasurface-based L2C polarization converter is proposed. The design not only features efficient polarization conversion, but also a flexible structure that can be applied in various curvature or conformal surfaces. Both the performances in normal and conformal situations are evaluated and discussed in this paper.

Due to the unprecedented circumstances brought about by the COVID-19 pandemic, the fabrication phase of my research design was unfortunately interrupted, resulting in its absence from the completed simulation results. The global pandemic and its associated restrictions significantly affected the operational capacity of laboratories and disrupted the supply chains necessary for prototype fabrication. Despite the challenges posed by the pandemic, the authors diligently conducted extensive simulation experiments as a substitute for the fabrication process. These simulations were designed to closely mirror the anticipated outcomes of the physical prototype. By leveraging state-of-the-art software and industry-standard simulation techniques, we were able to obtain valuable insights and assess the feasibility and performance of the proposed design.

While the absence of the physical prototype is regrettable, the simulation results present a robust foundation for evaluating the research design. They demonstrate the meticulous consideration given to the theoretical aspects, indicating a thorough understanding of the project’s objectives and potential outcomes.

## 2. Structure Design and Parametric Study

Theoretically, when an incident wave reaches the surface of the polarization converter, it will be reflected with a certain magnitude and phase. For instance, given a downward *y*-polarized incident wave, the reflected wave $E_{r}$ can be expressed by [1]:(1)$$E_{r}=E_{xr}{\hat{e}}_{x}+E_{yr}{\hat{e}}_{y}=E_{x}\exp\left(j\phi_{xy}\right){\hat{e}}_{x}+E_{y}\exp\left(j\phi_{yy}\right){\hat{e}}_{y},$$
where $E_{x}$ and $E_{y}$ are the *x* and *y* components of the electric field, respectively; ${\hat{e}}_{x}$ and ${\hat{e}}_{y}$ are the *x* and *y* unit direction vector, respectively; whereas $\phi_{xy}$ and $\phi_{yy}$ are their corresponding phases. From Equation (Equation 1), $R_{yy}$ and $R_{xy}$ are defined by Equations (Equation 2) and (Equation 3), respectively, to represent the reflection coefficient magnitudes of *y*-to-*x* and *y*-to-*y* polarization conversion, respectively,
(2)$$R_{xy}=\left|E_{xr}/E_{yi}\right|,$$
(3)$$R_{yy}=\left|E_{yr}/E_{yi}\right|.$$

To achieve an L2C polarization conversion, two conditions have to be fulfilled, which are
(4)$$R_{xy}=R_{yy}$$
and
(5)$$\Delta \phi =\phi_{yy}-\phi_{xy}=2n\pi \pm \pi /2,$$
where Δϕ is the phase difference between $E_{xr}$ and $E_{yr}$, and *n* is an integer. When Δϕ has a “−π/2” component, it indicates a right-hand circular polarization (RHCP), whereas a “+π/2” component indicates a left-hand circular polarization.

To further evaluate the polarization conversion, four Stroke parameters, *I*, *Q*, *U*, and *V*, are introduced, as shown in Equations (Equation 6)–(Equation 9) [1]:(6)$$I={\left|R_{xy}\right|}^{2}+{\left|R_{yy}\right|}^{2},$$
(7)$$Q={\left|R_{yy}\right|}^{2}-{\left|R_{xy}\right|}^{2},$$
(8)$$U=2\left|R_{xy}\right|\left|R_{yy}\right|\cos\Delta \phi ,$$
(9)$$V=2\left|R_{xy}\right|\left|R_{yy}\right|\sin\Delta \phi .$$

As the parameter to describe the degree to which the polarization has been converted, the ellipticity *e* is defined as
(10)e=V/I,
and it is a perfect LHCP conversion when *e* = 1, while it indicates a perfect RHCP conversion when *e* = −1.

Moreover, two angles α and β are defined by the Stroke parameters as expressed in Equations (Equation 11) and (Equation 12):(11)tan2α=U/Q,
(12)sin2β=V/I,
where α is the polarization azimuth angle while β is the ellipticity angle. Additionally, from the given phase difference Δϕ, the axial ratio (AR) can be calculated by [16]
(13)$$\mathrm{AR}={\left(\frac{{\left|R_{yy}\right|}^{2}+{\left|R_{xy}\right|}^{2}+\sqrt{a}}{{\left|R_{yy}\right|}^{2}+{\left|R_{xy}\right|}^{2}-\sqrt{a}}\right)}^{1/2},$$
where
(14)$$a={\left|R_{yy}\right|}^{4}+{\left|R_{xy}\right|}^{4}+2{\left|R_{yy}\right|}^{2}{\left|R_{xy}\right|}^{2}\cos\left(2\Delta \phi \right).$$

Another two parameters, energy conversion efficiency η and polarization conversion rate (PCR), are given by Equations (Equation 15) and (Equation 16), respectively,
(15)$$\eta =\left({\left|E_{xr}\right|}^{2}+{\left|E_{yr}\right|}^{2}\right)/{\left|E_{yi}\right|}^{2}={\left|R_{xy}\right|}^{2}+{\left|R_{yy}\right|}^{2}.$$
(16)$$\mathrm{PCR}=\frac{|R_{xy}{|}^{2}}{|R_{xy}{|}^{2}+{\left|R_{yy}\right|}^{2}}.$$

The proposed L2C polarization converter has a periodic unit cell, which combines an anisotropic “S”-shaped outer split ring resonator (SRR) and two “C”-shaped inner SRRs as shown in Figure 1a. The two “C”s are placed inside the openings of the “S” so that the entire structure is centrosymmetric. The proposed converter is designed on a flexible substrate, Rogers RT/duroid 5880, which has a thickness *t* = 1.575 mm and a dielectric constant $\varepsilon_{r}$ = 2.2. The periodicity of the unit cell is *p* = 13 mm.

The design is simulated in the ANSYS High-Frequency Structural Simulator (HFSS), where the unit cell is placed in an airbox as shown in Figure 1b and the distance between the unit cell and the top/bottom of the air box is set as λ/2, the half wavelength of the frequency of interest. To repeat the periodic structure, master and slave boundary conditions are set up for infinite array approximation. In addition, Floquet port is assigned to the top face of the radiation box to simulate the incident wave source, as shown in Figure 1b. The blue arrows represent the linearly polarized incident waves, while the red arrows represent the converted circularly polarized waves. Additionally, due to the coupling effects inside the proposed SRR, the induced current may be mainly distributed at the horizontal part of the “S”- and “C”-shaped structures.

For the optimization of the unit cell dimensions and postdesign tuning, several parametric studies are conducted to evaluate the influences caused by $g_{1}$ and $W_{s}$, namely the gap between “S” and “C” and the conductive trace width, respectively.

A parametric sweep for the gap between “S” and “C”, $g_{1}$, is set as values of 0.25 mm, 0.50 mm, 0.75 mm, 1.25 mm, 1.75 mm, and 2.25 mm, respectively. The impact of $g_{1}$ on the reflection coefficients of this design is described in Figure 2 and Figure 3. As can be seen in Figure 2a, the frequency of the *y*-to-*y* reflection gradually shifts from 12.2 GHz to 14.8 GHz with the increment in the value of $g_{1}$. In addition, $|R_{yy}|$, the magnitude of the *y*-to-*y* conversion reflection coefficient, reaches its maximum value of about 0.8 when $g_{1}$ is 1.75 mm and 2.25 mm, which means that 80% of the *y*-polarized incident wave is reflected, remaining the same polarization.

Correspondingly, there are also phase changes happening along with the reflection, which can be seen in Figure 2b. In the scenarios with different $g_{1}$ values, an abrupt 360° phase change (from −180° to 180°) can be observed at the same frequencies where the reflection happens.

On the other hand, the magnitude and phase of the reflection coefficient of the *y*-to-*x* polarization conversion $R_{xy}$ are reflected in Figure 3. Figure 3a presents the phenomenon of the frequency of reflection increasing from 12.4 GHz to 13.5 GHz as $g_{1}$ varies from 0.25 mm to 2.25 mm, which is similar to that of $R_{yy}$. The maximum of over 70% reflection happens at about 12.4 GHz when $g_{1}$ is 0.25 mm and 0.50 mm. As for the phase of *y*-to-*x* conversion, $\phi_{xy}$, it also has an abrupt change from −180° to 180° in each scenario.

Taking the phases of both *y*-to-*y* and *y*-to-*x* reflection from the above results, the phase difference Δϕ can be obtained as shown in Figure 4. According to the second condition of achieving L2C conversion in Equation (Equation 5), three gray dash reference lines are plotted in Figure 4 to indicate the values of interest when Δϕ = 90°, −90° and −270°. There are two regions where the Δϕ curves coincide with the reference dash lines: one is the region from 12.5 GHz to 14 GHz and Δϕ = −90°. For different values of $g_{1}$, the spans of the region vary significantly. Approximately, when $g_{1}=0.25$ mm, the curve has the largest span from 12.5 GHz to 14 GHz; when $g_{1}=0.75$ mm, the span is the narrowest near 13 GHz. The other region is from 14.1 GHz to 16 GHz with Δϕ = −270°, where each curve has roughly the same span.

In Figure 5a, the ellipticity *e* derived from the stroke parameters is illustrated. It can be observed that *e* reaches 1 in each case at a different frequency, which indicates the converted polarization is LHCP. No RHCP can be found for e=−1.

Another criterion to verify a circular polarization is the axial ratio, which is illustrated in Figure 5b. The highlighted green area indicates the <3 dB region for circular polarization, where only four curves can be found, i.e., $g_{1}$ = 0.25 mm, $g_{1}$ = 0.50 mm, $g_{1}$ = 0.75 mm, and $g_{1}$ = 1.25 mm.

The energy conversion efficiency η and polarization conversion rate are illustrated in Figure 6a and Figure 6b, respectively. It is clear that the energy conversion efficiency of each case is above 80%, except for the case of $g_{1}$ = 1.25 mm and $g_{1}$ = 2.25 mm. On the other hand, the maximum PCRs gradually decrease as $g_{1}$ increases. When $g_{1}$ = 0.25 mm, it has the highest maximum PCR of about 58%.

Next, a parametric sweep of the trace width $W_{s}$ is simulated, where $W_{s}$ is set as 0.25 mm, 0.45 mm, 0.50 mm, 0.65 mm, and 0.75 mm. The impacts of $W_{s}$ on the *y*-to-*y* reflection component are illustrated in Figure 7. As can be seen in Figure 7a, the magnitude of the *y*-to-*y* reflection has a minimum at a certain frequency, which shifts from 11.8 GHz to 13 GHz as $W_{s}$ increases. When $W_{s}$ is 0.25 mm, $|R_{yy}|$ is 0.6, which means that 60% of the *y*-polarized incident wave is reflected as a *y*-polarized wave. Correspondingly, abrupt 360° changes in phase $\phi_{yy}$ can be found in Figure 7b and the frequencies where it happens also have the same shift with the magnitude minimum.

As for the impact of $W_{s}$ on the *y*-to-*x* reflection, it is presented in Figure 8. It shows clearly that the peaks of $|R_{xy}|$ shift from 11.7 GHz to 12.8 GHz as the value of $W_{s}$ increases in Figure 8a. All peak values of $|R_{xy}|$ are above 0.65, which means that, at those frequencies, more than 65% of the *y*-polarized waves are reflected as *x*-polarized ones. The reflections can also be verified by the 360° change in $\phi_{xy}$ in Figure 8b. It is also noticeable that the phase reverse happens from 12 GHz to 14 GHz, after which the phase curves continue along with their original tendency.

Combining both $\phi_{yy}$ in Figure 7b and $\phi_{xy}$ in Figure 8b, the phase difference δϕ is presented in Figure 9a. According to the reference dash lines, two regions where Δϕ is −270° and −90° can be found at 12.1–14 GHz and 14–16 GHz, respectively. The span of each curve in the first region decreases as $W_{s}$ increases, whereas each curve has nearly the same span in the latter region. Apart from these regions, there are also two points on each curve where Δϕ = 90°.

Figure 9b shows the ellipticity *e* for the parametric sweep of $W_{s}$. One noticeable result is that each scenario has the point where *e* = 1, indicating LHCP. In addition, the ellipticity is approaching −1 at 12.8 GHz when $W_{s}$ = 0.75 mm, which is a tendency that an RHCP can achieve at some point.

The effect of varying $W_{s}$ on the axial ratio is presented in Figure 10 and the highlighted green region suggests that the <3 dB part is qualified as circular polarization. It can be observed that each curve has a different portion within the highlighted green region, among which it reaches the maximum at 12 GHz when $W_{s}$ is 0.25 mm.

The effects of $W_{s}$ on the energy conversion efficiency η and PCR are illustrated in Figure 11a and Figure 11b, respectively. It is clear that over 90% of energy conversion efficiency is achieved for different $W_{s}$ values in Figure 11a. On the other hand, the overall PCR is above 50% for each scenario in Figure 11b and the frequency of the maximum conversion rate shifts from 11.8 GHz to 13 GHz as the value of $W_{s}$ increases.

From all parametric sweep results of both the gap between the “S” and “C” ($g_{1}$) and the trace width ($W_{s}$), significant shifts along the frequency can be found in the magnitude and phase of the two reflection and conversion components when the parameter value changes. Thus, this is solid proof that the operating frequency is tunable by adjusting the value of those parameters. With the current sweep range, the converter can be tuned between 12 GHz and 14 GHz. In addition, both the energy conversion efficiency and polarization conversion rate are relatively high during the sweep simulation.

## 3. Results and Discussion

After the parametric study and optimization in HFSS, one set of parameters is chosen so that the proposed design can achieve L2C polarization conversion at 12.4 GHz. The finalized dimensional parameters are listed as *p* = 13 mm, $d_{1}$ = 5.75 mm, $d_{2}$ = 4.25 mm, $d_{3}$ = 8 mm, *w* = 0.5 mm, $g_{1}$ = 0.75 mm, and $g_{2}$ = 0.5 mm, as shown in Figure 1a.

According to Equation (Equation 4), at 12.4 GHz, the magnitudes of the reflection coefficients $R_{yy}$ and $R_{xy}$ need to be equal. As shown in Figure 12a, $|R_{yy}|$ and $|R_{xy}|$ are approximately equal with a value of 0.70 as highlighted in the yellow area. Moreover, a phase difference of Δϕ = −270° (or 90°) can be found on the pink curve in Figure 12b, which suggests that the reflected wave is an LHCP wave. This is also verified by the value 1 of ellipticity *e* at the frequency of interest as shown on the blue curve.

In addition, as illustrated in Figure 12b, the AR of the proposed design is below 3 dB (highlighted in yellow), which indicates that the final electromagnetic wave is qualified as circular polarization. It also suggests that the energy conversion efficiency of this design is about 92% at 12.4 GHz.

To testify the enhancement of the L2C converter, a microstrip patch antenna operating at 12.4 GHz is placed under a finite 4 × 4 unit cell array as shown in Figure 13. The distance between the patch antenna and the L2C converter is a quarter wavelength λ/4 = 4 mm in the dielectric substrate at 12.4 GHz.

As shown in Figure 13, it can be seen clearly that the bandwidth of the patch antenna without applying the L2C converter is 0.71 GHz (from 12.13 GHz to 12.84 GHz), whereas the bandwidth of the patch antenna with the L2C converter increases to 1.43 GHz as highlighted in green color (from 11.94 GHz to 13.37 GHz), which is nearly doubled.

## 4. Conformal Test with Wavy Curvature Surface

To be applied to conformal surfaces and human body wearable applications, the polarization converter is proposed on a flexible substrate. In this section, a conformal test is conducted in HFSS to evaluate the performance and tolerance of the proposed polarization converter.

In the test, the entire unit cell is bent along a sinusoidal curve on the XoZ-plane, where the period of the sine wave equals the periodicity of the unit cell, i.e., T=p=13 mm. Thus, the curvature of the structure is also repeated in the unit cell. In this case, the amplitude of the sine wave is set as a variable to simulate different levels of bending, namely $A_{bend}$. The value of $A_{bend}$ is swept from 0 mm to 1 mm with a step size of 0.25 mm.

The magnitude and phase change of the *y*-to-*y* reflection coefficient $R_{yy}$ are illustrated in Figure 14, and those of the *y*-to-*x* reflection coefficient $R_{xy}$ are presented in Figure 15. It can be observed that, for both $R_{yy}$ and $R_{xy}$, their peak magnitudes have minor shifts from 12 GHz to 13 GHz as $A_{bend}$ increases from 0 mm to 1 mm gradually. This suggests that the frequency where reflection conversion happens has minor deviations due to the conformal surface.

As the second condition to achieve L2C polarization conversion, the phase difference Δϕ is important. The impact of the sinusoidal conformal surface on it is presented in Figure 16, from which it is noticeable that there are only two points where Δϕ is −270° between 12 GHz and 14 GHz for each case except the one without bending ($A_{bend}$ = 0 mm).

On the other hand, the results of ellipticity *e* and axial ratio for the conformal bending test are given in Figure 17a and Figure 17b, respectively. The frequencies wherein *e* = 1 and AR < 3 dB can be found in the figures with minor shifts, in agreement with those in Figure 14 and Figure 15.

The overall results of the conformal tests suggest that the proposed L2C polarization converter is subject to an operating frequency shift due to the sinusoidal conformal surface. In addition, the phase difference has no perfect range that matches for ±90°, but only two intersection points. This suggests that the two converted components may not be in perfect ±90° phase difference, which will lead to an imperfect circular polarization.

## 5. Conclusions

A metasurface-based L2C polarization converter is proposed to convert a linearly polarized incident wave into an LHCP wave so that the return loss bandwidth can be broadened. The L2C polarization converter has a low-profile single-layer structure and operates at 12.4 GHz. It consists of a nested “S”- and “C”-shaped SRR structure. Parametric studies and simulation results are given for the design, which show that, at the desired frequency, the converter has a high energy conversion efficiency and polarization conversion rate. Further conformal tests are conducted to testify the performance of the design when applied to flexible and curvature surfaces. The result shows that it is subject to a minor operating frequency shift.

## Figures and Tables

**Figure 1 sensors-23-06268-f001:**
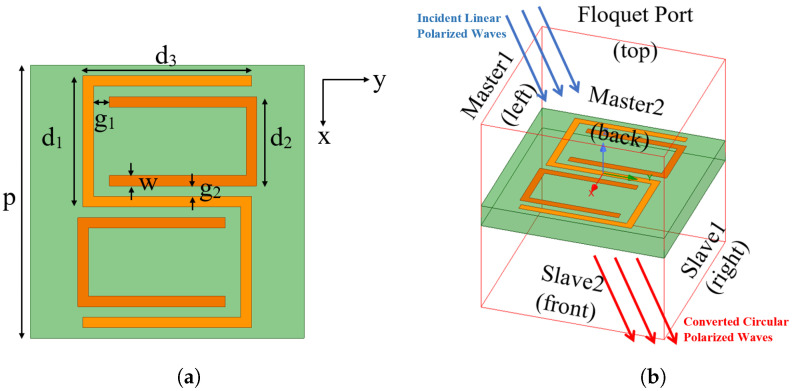
Antenna layout and simulation setup; (**a**) Antenna layout and dimensions; (**b**) Simulation setup in HFSS.

**Figure 2 sensors-23-06268-f002:**
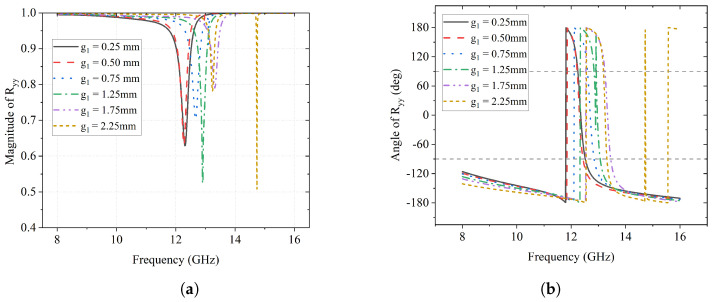
Effect of $g_{1}$ on $R_{yy}$ (**a**); Magnitude of $R_{yy}$ (**b**); Angle of $R_{yy}$ (deg).

**Figure 3 sensors-23-06268-f003:**
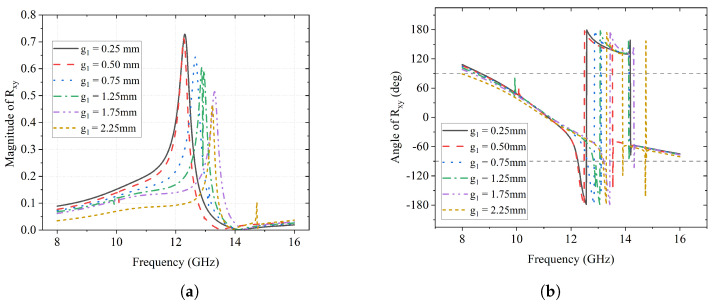
Effect of $g_{1}$ on $R_{xy}$ (**a**); Magnitude of $R_{xy}$ (**b**); Angle of $R_{xy}$ (deg).

**Figure 4 sensors-23-06268-f004:**
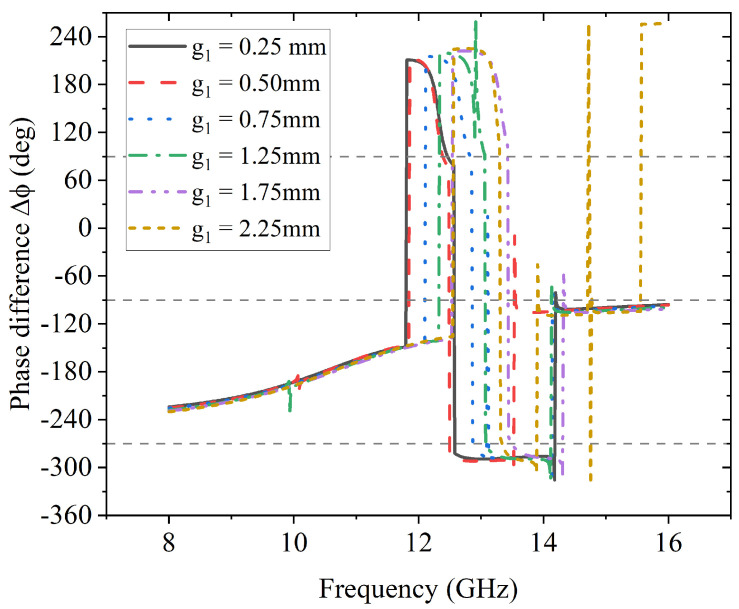
Effects of $g_{1}$ on phase difference Δϕ.

**Figure 5 sensors-23-06268-f005:**
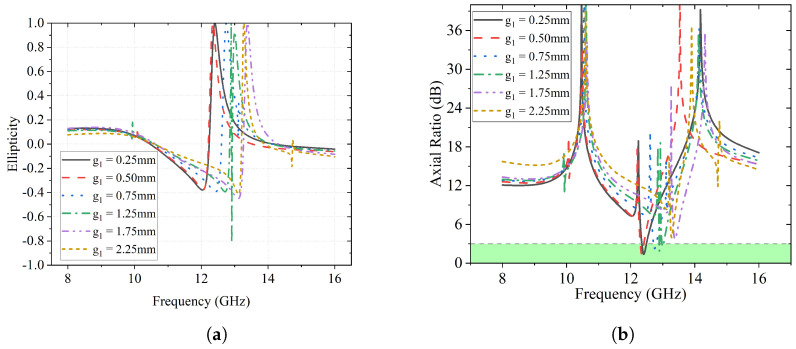
Effects of $g_{1}$ on ellipticity *e* and AR. (**a**) Effects of $g_{1}$ on ellipticity *e*. (**b**) Effects of $g_{1}$ on axial ratio.

**Figure 6 sensors-23-06268-f006:**
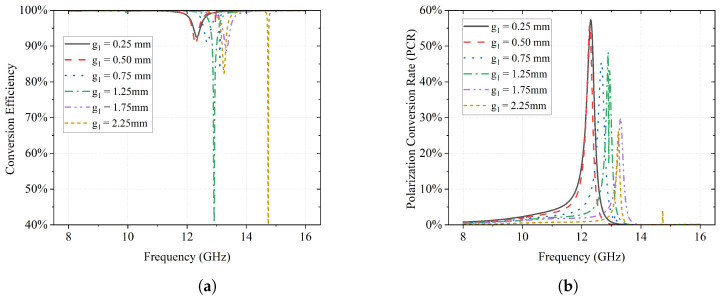
Effects of $g_{1}$ on conversion efficiency η and PCR. (**a**) Effects of $g_{1}$ on conversion efficiency η. (**b**) Effects of $g_{1}$ on polarization conversion rate.

**Figure 7 sensors-23-06268-f007:**
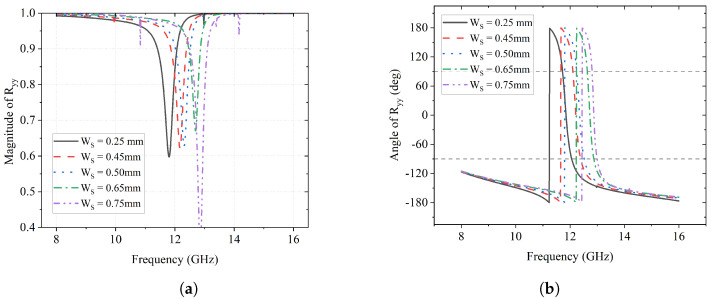
Effect of $W_{s}$ on $R_{yy}$. (**a**) Magnitude of $R_{yy}$; (**b**) Angle of $R_{yy}$ (deg).

**Figure 8 sensors-23-06268-f008:**
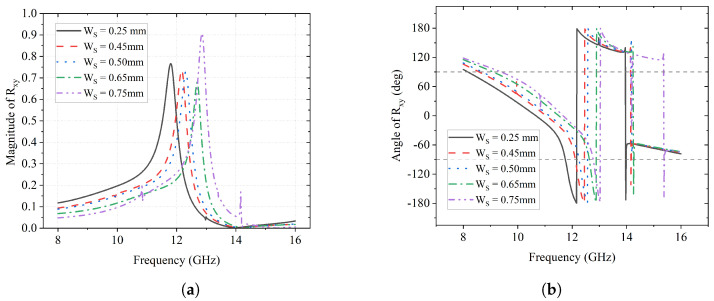
Effectof $W_{s}$ on $R_{xy}$. (**a**) Magnitude of $R_{xy}$; (**b**) Angle of $R_{xy}$ (deg).

**Figure 9 sensors-23-06268-f009:**
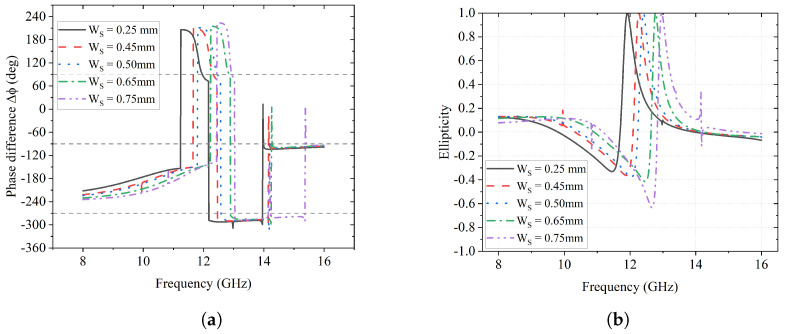
Effects of $W_{s}$ on phase difference Δϕ and ellipticity *e*. (**a**) Effects of $W_{s}$ on phase difference Δϕ; (**b**) Effects of $W_{s}$ on ellipticity *e*.

**Figure 10 sensors-23-06268-f010:**
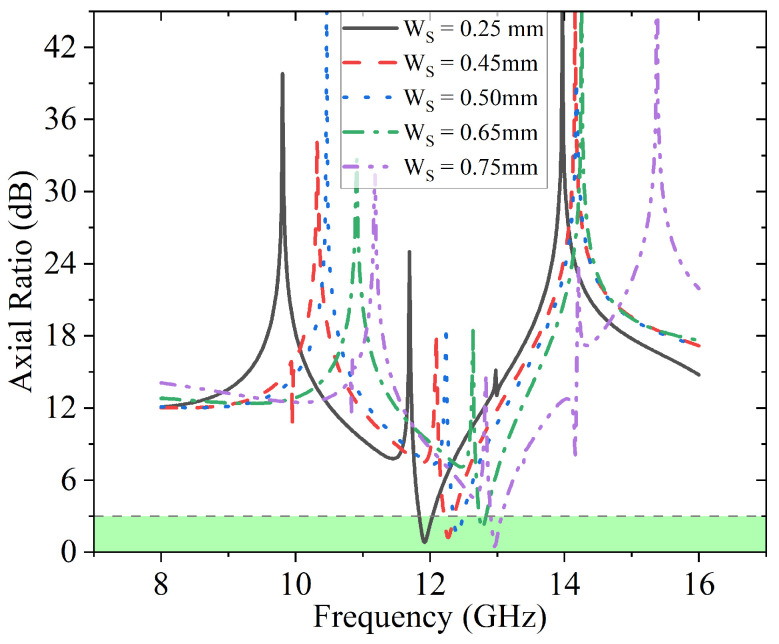
Effects of $W_{s}$ on axial ratio.

**Figure 11 sensors-23-06268-f011:**
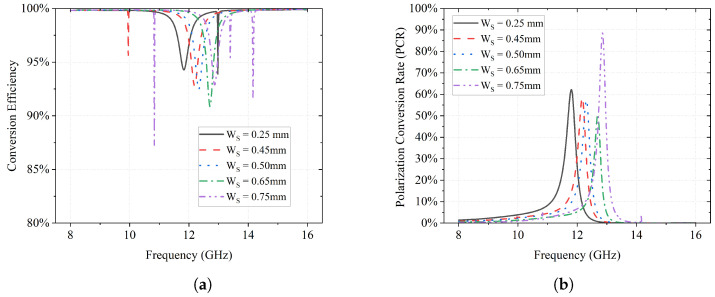
Effects of $W_{s}$ on conversion efficiency η and PCR. (**a**) Effects of $W_{s}$ on conversion efficiency η; (**b**) Effects of $W_{s}$ on PCR.

**Figure 12 sensors-23-06268-f012:**
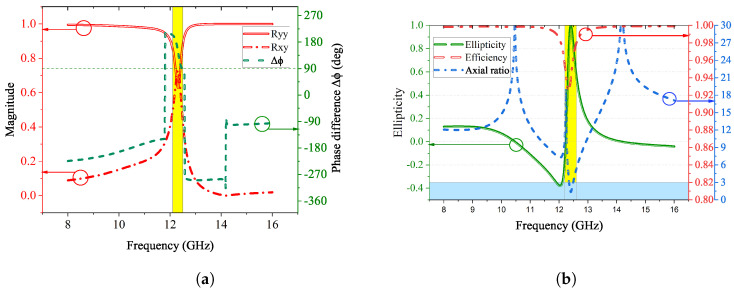
Performance of proposed L2C polarization converter. (**a**) Magnitudes and phase difference of reflection coefficients; (**b**) Ellipticity, efficiency, and axial ratio of the L2C converter.

**Figure 13 sensors-23-06268-f013:**
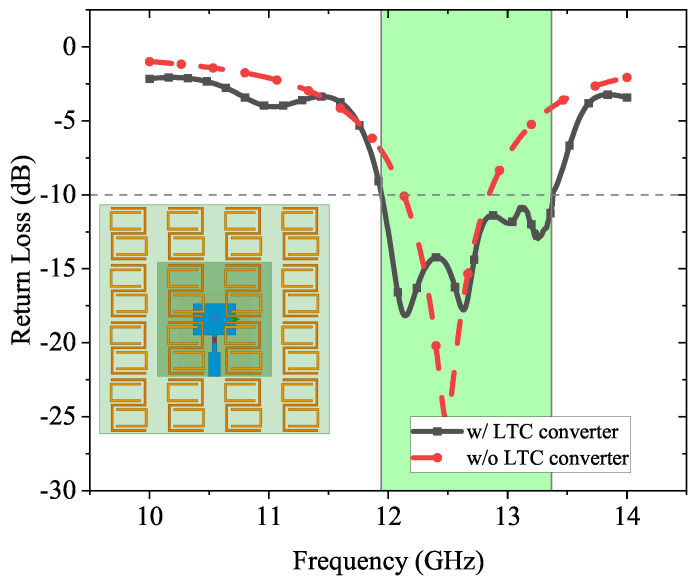
Simulation setup and results of the patch antenna and proposed L2C converter.

**Figure 14 sensors-23-06268-f014:**
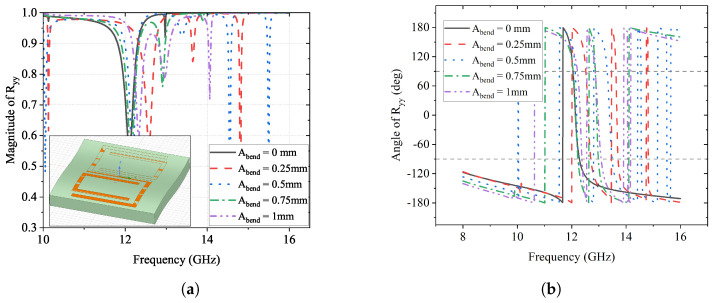
Effects of $A_{bend}$ on $R_{yy}$. (**a**) Magnitude of $R_{yy}$; (**b**) Angle of $R_{yy}$ (deg).

**Figure 15 sensors-23-06268-f015:**
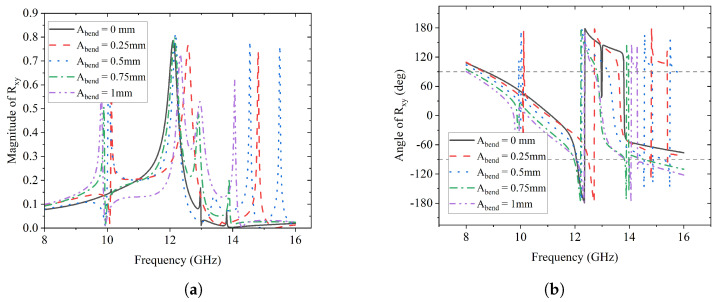
Effects of $A_{bend}$ on $R_{xy}$. (**a**) Magnitude of $R_{xy}$; (**b**) Angle of $R_{xy}$ (deg).

**Figure 16 sensors-23-06268-f016:**
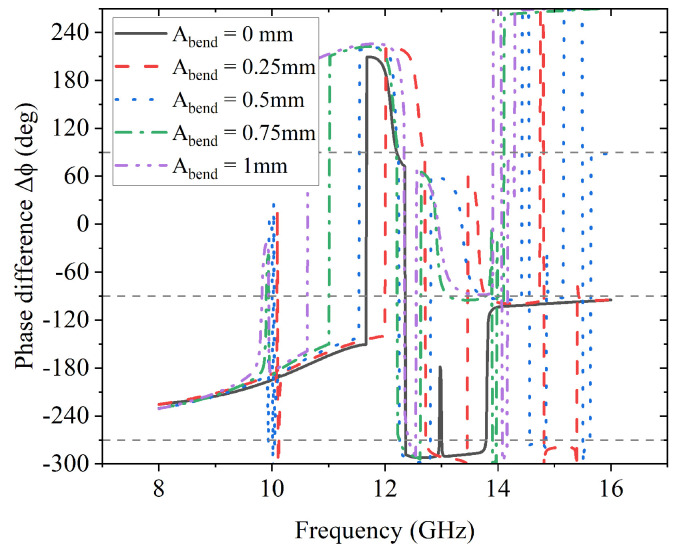
Effects of $A_{bend}$ on phase difference Δϕ.

**Figure 17 sensors-23-06268-f017:**
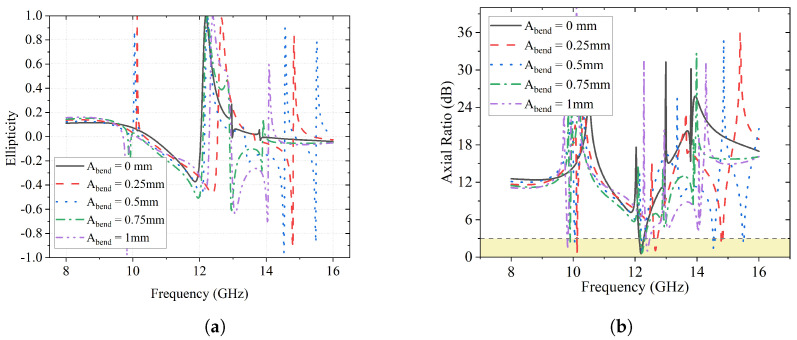
Effects of $A_{bend}$ on ellipticity *e* and AR. (**a**) Effects of $A_{bend}$ on ellipticity *e*; (**b**) Effects of $A_{bend}$ on axial ratio.

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
