# Peer review of "A Metasurface-Based LTC Polarization Converter with S-Shaped Split Ring Resonator Structure for Flexible Applications"

_sensors, 2023, doi:10.3390/s23146268_

Round 1

Reviewer 1 Report

The manuscript proposes a metasurface structure for linear to circular polarization conversion, based on metallic split ring resonators. Simulation results show that it has high conversion efficiency at 12.4 GHz. It is also demonstrated by simulations that when there is a curvature on the substrate, the resonant peak will be shifted. My comments are listed as follows:

1.     The novelty of this work is not high. So far, a large amount of metasurface-based polarization converts has been proposed in the GHz band. Many of them have not only linear to circular polarization converting function, but also multi-functions, or they have broadband performance and low incident angle dependence. This manuscript just has one function in one frequency, which do not have advantages compared with the others. Although the study on the curved substrate may be a little bit novel, this is only a very small part of the whole manuscript.

2.     Compared with the other similar works, this work is only a simulation study. In my opinion, the fabrication of a metallic metasurface for GHz waves is not difficult. The authors should have an experimental demonstration.

3.     There are only simulation results shown in the manuscript, no any physical explanations on the results such like why the peak shifts. This does not meet the criteria of a research paper.

4.     I also suggest the authors give an incident angle independent performance of the device.

Therefore, I do not suggest the manuscript be accepted by Sensors.

The Engligh writing is poor, including:

a)       gramma mistakes.

b)      spelling mistakes, such as ‘Stroke parameters’.

c)       unspecified abbreviation, such as FSS.

d)      undefined parameters, such as Eyi in Equation (2).

Author Response

Please kindly refer to the uploaded document. Thank you very much.

Reviewer 2 Report

1) The abbreviation LTC used is not appropriate. It is often used for Low Temperature Ceramics. In this case, it is confusing. I suggest L2C or L-C.

2) Er& in eq.1 is not explained. Mainly what the & symbol means. 

3) The same symbol e is used for two different quantities. e in eq.1 probably means something different than in eq.10.

4) In line 97, it should not be Rogers RT/droid 5880, but Rogers RT/duroid 5880.

5) If the thickness of the Duroid substrate is t = 1.575 mm, then it cannot be considered a flexible substrate as stated in the title of the article or in other places. Flexible can be significantly thinner substrates <0.15 mm.

6)The readability of Fig. 2, 3, 5, 6, 7, 8, 9, 11, 12, 14, 15, 17 is impaired by reducing the graphs.

Author Response

(The authors gave the same response as above.)

Reviewer 3 Report

The author proposed S-Shaped and Split Ring Resonator (SRR) metamaterials to convert the linear-to-circular (LTC) polarization. The authors demonstrate that their metamaterial can convert the linearly polarized incident wave into the left-handed circularly polarized wave at a frequency of 12.4 GHz. However, this work lacks experimental results, and similar work has been published [Fourth International Conference on Electrical, Computer and Communication Technologies (ICECCT), pp. 1-3. IEEE, 2021]. Therefore, I recommend that the manuscript be accepted after undergoing major revisions. My comments are detailed below:

1.      This work demonstrates the point of view of simulation-based. However, the validity of the simulation results is quite doubtful if no experimental results are presented. How did the author overcome this comment? The author should put some reason or discussion in the manuscripts.

2.      Similar previous works that present the metamaterials-based LTC polarization converter with an experimental approach have been published in [Scientific reports 10, no. 1 (2020): 17981], [JOSA B 35, no. 4 (2018): 950-957], [Scientific Reports 11, no. 1 (2021): 9306] and [Scientific reports 9, no. 1 (2019): 4552]. Apart from the proposed design of LTC metamaterial based on an S-Shaped and Split Ring Resonator (SRR) structure, what is the novelty of this work? The author should put more discussion in the introduction.

3.      The proposed metamaterial can transform the linearly polarized into circularly polarized. However, is the proposed metamaterial reversible, such as transforming the circularly polarized into linearly polarized?

4.      Could the author compare the performances of the proposed LTC metamaterial to the previous LTC metamaterials, such as the bandwidth, polarization conversion ratio, and angular stability?

5.      The schematical drawing of LTC metamaterial in Figure 1 lacks the three-dimensional cartesian coordinate axes. Moreover, the schematical drawing of the excitation of waves is not demonstrated clearly.

6.      The figures in this manuscript have less explanation, such as Figures 3(b), 14(a), 14(b), 15(a), and 15(b). Furthermore, the author did not explain the second peak of g1 = 2.25 mm in Figures 2(a) and 2(b); multiple peaks on Ws = 0.75 mm in Figure 11(a); multiple peaks at A bend of 0.25 mm, 0.5 mm, 1mm in Figure 14(a).

7.      The manuscript mentioned flexible applications in the manuscript but lacked schematical drawing of bending. Moreover, why does the author present the sine wave to demonstrate the bending? The simulation commonly uses a bending angle or radius as the bending parameter.

8.      To help the readers have a more comprehensive understanding of the new metamaterial, I suggest supplementing some latest works about ultrastrong electromagnetic resonance [Photonics Research 10, no. 9 (2022): 2215-2222], conversion between polarization states based on a metasurface [Photonics Research, 7(3), pp.246-250], multifunctional all-dielectric metasurface quarter-wave plates for polarization conversion and wavefront shaping [Optics Letters 47, no. 10 (2022): 2478-2481], and terahertz transmissive half-wave metasurface with enhanced bandwidth [Optics Letters, 46(17), pp.4164-4167]. 

Extensive editing of the English language required

Author Response

(The authors gave the same response as above.)

Reviewer 4 Report

11 May 2023

Review of the manuscript 2297817

“A Metasurface Based LTC Polarization Converter with S-Shaped Split Ring Resonator Structure for Flexible Applications”

 I am interested about the metasurface LTC polarization converter and its properties. I have some questions what I want to know as follows.

1. Page 4, line 99-102, Figure 1: Author indicates “the top/bottom of the air box is set as λ/2, the half wavelength of the frequency of interest.” Please show the propagation direction of the wave by using a schematic figure overwriting on Figure 1. 

2. Page 4, line 109-116, Author indicated detail sizes and structure of LTC unite cell which is used in this research. Please explain the principle why the linear polarization wave can be converted to circular polarization wave, if it possible.

3. Figure 2, 3, 4, 5 and 6; Please explain why the peaks move with respect to the size of g1.  How about the effect of g2 are?

4. Page 8, line 210-213:  Author mentioned the benefit of the LTC by the bandwidth (from 11.94 to 13.37 GHz).  Could it be used as an antenna for satellite communications? Because, in Japan, the broadcasting satellite (BS) use 11.7-12.2 GHz band and the communications satellite (CS) uses 12.2-12.75 GHz band.

5.  Page 9, line 240-244: As a future task, author should check that the LTC can control the direction of rotation the circular polarization vector to right-hand or left-hand.

 These five are my questions.

Author Response

(The authors gave the same response as above.)

Round 2

Reviewer 1 Report

The authors has improved the manuscript accordingly. However, I still believe experimental results are needed.

Reviewer 3 Report

The author has great responses to the reviewer’s comments. I appreciated the authors’ effort in modifying their manuscript. However, there are several comments for the revised manuscript.

1. This manuscript should present a discussion of nearfields and current distributions.  

2. The discussion of the Q factor and FOM should be better presented.

3. The statement, “The author’s intention was to fabricate prototypes of the proposed design and measure the results to compare with the simulation. However, this was delayed due to various reasons during the pandemic. Thus, fabrication is our next step to enrich and enhance the results” seems weak. The author should improve their reason and put it into the introduction section.

Considering several comments above, I would like to accept after minor revision this manuscript.
